

# Undergraduate data science degrees emphasize computer science and statistics but fall short in ethics training and domain-specific context

Jeffrey C. Oliver[1] and Torbet McNeil[1,2]

[1] Office of Digital Innovation & Stewardship, University Libraries, University of Arizona, Tucson, AZ, USA
[2] Department of Educational Policy Studies and Practice, University of Arizona, Tucson, AZ, USA

## ABSTRACT

The interdisciplinary field of data science, which applies techniques from computer science and statistics to address questions across domains, has enjoyed recent considerable growth and interest. This emergence also extends to undergraduate education, whereby a growing number of institutions now offer degree programs in data science. However, there is considerable variation in what the field actually entails and, by extension, differences in how undergraduate programs prepare students for data-intensive careers. We used two seminal frameworks for data science education to evaluate undergraduate data science programs at a subset of 4-year institutions in the United States; developing and applying a rubric, we assessed how well each program met the guidelines of each of the frameworks. Most programs scored high in statistics and computer science and low in domain-specific education, ethics, and areas of communication. Moreover, the academic unit administering the degree program significantly influenced the course-load distribution of computer science and statistics/mathematics courses. We conclude that current data science undergraduate programs provide solid grounding in computational and statistical approaches, yet may not deliver sufficient context in terms of domain knowledge and ethical considerations necessary for appropriate data science applications. Additional refinement of the expectations for undergraduate data science education is warranted.

# BACKGROUND

Data-intensive work and the desire for data-driven decisions increasingly fuel interest in the field of data science. According to the *National Academies of Sciences, Engineering & Medicine (2018)*, employers across disciplines are demanding that employees have skills in working with and extracting knowledge from data. Additionally, the report suggests every undergraduate student should graduate with at least beginning competency for working with data. However, many undergraduates are not obtaining the necessary training to prosper in the new economy. College administrators have reacted, and the number of undergraduate data science programs has multiplied. In 2014, there were

Corresponding author
Jeffrey C. Oliver,
jcoliver@email.arizona.edu

approximately 13 undergraduate major programs (*Aasheim et al., 2015*), while at least 50 undergraduate programs existed in the United States as of September 2020 (*Swanstrom, 2020*). Interest in data science in undergraduate education is also evidenced by the growth of courses such as the University of California, Berkeley's lower-division Foundations of Data Science course, which increased from 100 students in 2013 to over 1,000 students in 2018 (*Kafka, 2018*). Similar growth can be expected internationally, particularly in more developed, knowledge-based economies striving to improve their universities, as the U.S. higher education model is often emulated (*Bok, 2013*).

Despite the attention data science is receiving, there is a lack of a clear definition of what data science actually entails (*Donoho, 2017*; *Irizarry, 2020*). While an exhaustive history of the term and definitions is beyond the scope of this work, we adhere to the following general description: Data science draws on statistics and computer science to address questions in various domains, such as biology, education, physics, business, linguistics, or medicine (*Donoho, 2017*; *National Academies of Sciences, Engineering & Medicine, 2018*). In addition to quantitative skills and domain expertise, this definition of data science also includes "soft skills," such as communication and a working cognizance of the ethics of data use and reuse (*Irizarry, 2020*). This definition of data science is relatively broad and includes the narrower fields of data engineering (development and management of data infrastructure for subsequent interrogation and analyses) and data analytics (application of statistical and predictive analyses to address unknowns in a particular domain). The training students receive when pursuing a data science education, both what is included and what is excluded, is worthy of investigation.

While data science undergraduate degree programs are relatively new, previous work has highlighted early trends. In a survey of five data science degree programs, *Aasheim et al. (2015)* found an emphasis on statistics and mathematics coursework. All programs required coursework in linear algebra and data mining as well as multiple programing and statistics courses. In contrast, none of the five programs required coursework in ethical considerations of data science, and only one program required coursework in communication skills. Additional descriptions of individual data science undergraduate programs are quite varied and include those that are business-focused (*Anderson, McGuffee & Uminsky, 2014*) and those that require significant coursework in domains outside of mathematics, statistics, and computer science (*Anderson et al., 2014*). The variation among programs illustrates a potentially confusing landscape for students to navigate and uncertainty for employers when assessing recent graduates' preparation (*Parry, 2018*).

Quantitative evaluation of data science programs requires an explicit framework describing the components of data science education. Several frameworks exist, including general frameworks focused on mathematical and computational foundations (*De Veaux et al., 2017*), frameworks based on individual programs (*Anderson et al., 2014*), and emergent frameworks developed for comparative analyses (*Aasheim et al., 2015*). For this work, we used two frameworks for evaluation: the broad, explicit framework presented in the *National Academies of Sciences, Engineering & Medicine (2018)* report and the narrower, more conceptual framework of *Donoho (2017)*. The framework of the *National*

**Table 1 Ten areas of emphasis in the NASEM framework.**

Mathematical foundations

Computational foundations

Statistical foundations

Data management and curation

Data description and visualization

Data modeling and assessment

Workflow and reproducibility

Communication and teamwork

Domain-specific considerations

Ethical problem solving

**Table 2 Six areas of emphasis in the GDS framework.**

Data gathering, preparation, and exploration

Data representation and transformation

Computing with data

Data modeling

Data visualization and presentation

Science about data science

*Academies of Sciences, Engineering & Medicine (2018)*, hereafter referred to as the National Academies of Sciences, Engineering, and Medicine (NASEM) framework, focuses on developing undergraduate data acumen through a curriculum including important data science concepts, applications to real-world problems with an understanding of limitations, and ethical concerns involved in data science (Table 1; Table S1). In addition to "traditional" data science competencies in computer science and statistics, this framework gives guidelines for training in communication skills, domain-specific knowledge, and ethical considerations. The NASEM framework lists ten key concept areas ("areas" hereafter) that are further divided into specialized topics, skills, or concepts ("sub-areas" hereafter), providing comprehensive expectations for undergraduate training. This framework facilitates an extensive evaluation of how well undergraduate degree programs meet the expectations set forth by the NASEM.

The framework presented by *Donoho (2017)* as Greater Data Science, hereafter the GDS framework, describes programs that prepare professionals for gaining insights from data while applying best practices (Table 2; Table S2). The six areas in the GDS framework are relatively high-level, and the framework has notably little discussion of the knowledge, skills, or abilities necessary to apply such foundational knowledge to domain-specific questions. While not explicitly designed for undergraduate education, the GDS framework furnishes a metric to assess undergraduate programs' potential for preparing future professionals in data science careers.

Here we take the opportunity to evaluate undergraduate data science degree programs in a comparative analysis using the two frameworks described above. Applying an evaluation rubric we developed for each of the two frameworks, we investigated data science programs from a sample of doctoral-granting universities. We reviewed major requirements and corresponding course descriptions to assess how well each program addressed elements of each framework. We also quantified the relative amount of coursework in three categories: computer science, statistics/mathematics, and domain knowledge. Using evaluations and quantification of coursework, we provide an overview of how well each of the two frameworks is being implemented and an evaluation of training strengths and weaknesses in data science undergraduate degree programs.

## METHODS

In an attempt to make appropriate comparisons among undergraduate data science programs, we chose institutions comparable to our home institution, the University of Arizona. We used two means of inclusion: institutions recognized as peers by the University of Arizona and institutions in the Pac-12 Conference (https://uair.arizona.edu/content/ua-peers). The union of University of Arizona peers and Pac-12 institutions resulted in a total of 25 universities, all of which are Research I universities (*The Carnegie Classification of Institutions of Higher Education, 2018*) and 4-year, doctoral-granting institutions. While half of these 25 institutions are in the western United States, this sample also includes public universities in the midwestern, eastern, and southern United States. This sample includes members of the Association of American Universities (https://www.aau.edu/) and land-grant institutions. At the time of inception of this work, roughly 50% of these institutions (53% of University of Arizona peer institutions and 50% of Pac-12 institutions) offered an undergraduate major or minor in a data science-related field. A total of 10 institutions had at least one undergraduate major in data science. Variation in the names of programs required careful consideration. For example, Ohio State University offered a bachelor of science in data analytics; however, the curriculum was similar to programs with the term "data science" in the name of the degree, so this program was included in our evaluation. In contrast, the bachelor of science in business data analytics at Arizona State University had a course curriculum that was very different from other data science degrees and was, thus, excluded from evaluation. In cases when an institution offered more than one degree in data science (e.g., University of Washington), we scored each of the degree programs independently. This selection process resulted in a total of 18 scored programs (Table 3).

We evaluated how well each undergraduate data science program aligned with recommendations in the NASEM and GDS frameworks. For each of the two frameworks, we developed a rubric and coded the undergraduate data science curricula on a four-point scale using direct survey methodology, specifically content analysis of course descriptions (*Stefanidis & Fitzgerald, 2014*; *Aasheim et al., 2015*), indicating the familiarity with a topic that could be expected from a student graduating from the program in question. The creation of the rubric used for scoring was an iterative process. We created an initial rubric and used it to score each item in the two frameworks. For nine of the ten areas of the

**Table 3 Undergraduate data science programs considered in this work.** Administering unit lists the academic unit(s) responsible for the major, minor, or certificate program.

| Institution | Program | Administering unit |
|---|---|---|
| Ohio State University - Main Campus | B.S. in Data Analytics | Computer Science & Engineering; Statistics |
| Pennsylvania State University - Main Campus | B.S. in Data Sciences, Applied Data Science Option | Information |
| Pennsylvania State University - Main Campus | B.S. in Data Sciences, Computational Science Option | Computer Science & Engineering |
| Stanford University | Minor in Data Science | Statistics |
| University of Arizona | B.A. in Statistics and Data Science | Mathematics |
| University of Arizona | B.S. in Statistics and Data Science | Mathematics |
| University of California - Berkeley | B.A. in Data Science | Data Science |
| University of California - Davis | B.S. in Statistics, Statistical Data Science Track | Statistics |
| University of Colorado - Boulder | B.A. in Statistics and Data Science | Mathematics |
| University of Illinois at Urbana-Champaign | Certificate in Data Science | Statistics |
| University of Iowa | B.S. in Data Science | Statistics & Actuarial Science |
| University of Maryland - College Park | B.S. in Computer Science, Data Science Specialization | Computer Science |
| University of Washington - Seattle | B.S. in Applied & Computational Mathematical Sciences, Data Sciences & Statistics Track | Applied & Computational Mathematical Sciences |
| University of Washington - Seattle | B.S. in Computer Science, Data Science Option | Computer Science & Engineering |
| University of Washington - Seattle | B.S. in Human Centered Design & Engineering, Data Science Option | Human Centered Design & Engineering |
| University of Washington - Seattle | B.S. in Informatics, Data Science Track | Information |
| University of Washington - Seattle | B.S. in Statistics, Data Science Option | Statistics |
| Washington State University | B.S. in Data Analytics | Computer Science & Engineering; Mathematics & Statistics |

NASEM framework, an item corresponded to one sub-area within the larger area. For example, in the area of computational foundations, five sub-areas were listed: basic abstractions, algorithmic thinking, programing concepts, data structures, and simulations. The ability of a program to address each of these five sub-areas was assessed separately. One area in the NASEM framework, domain-specific considerations, did not list any additional specifics; so, in this case, the item scored was the area of domain-specific considerations itself. Similarly, in the GDS framework, most areas lacked additional descriptions of sub-areas, so items largely corresponded to the particular area. The two exceptions were data representation and transformation and data modeling; each of these areas had two scored sub-areas.

In general, a score of "1" indicated no expectation that a student graduating from the program would have familiarity with the area/sub-area; a score of "4" indicated a student is well-versed in the area/sub-area, with at least one required course covering the topic, often at length. Scores were based on course titles and descriptions only (links to each programs' web page are available in Table S3 and lists of course requirements are

available in Supplemental File 1); a lack of standardization among institutions required close reading of all course descriptions listed in posted curricula. We did not include information from course syllabi as there was considerable variation in which courses had publicly available syllabi. There was considerable revision to the rubric throughout the coding process as nuances in scoring and inconsistencies were noticed. We scored each program for each framework independently and then discussed discrepancies to reach agreements on a common score. Full details of the final coding rubric are available in Appendix A.

In addition to scoring programs for the two frameworks, we quantified the total number of credits required for each program in three categories: computer science, statistics/mathematics, and domain-specific courses. Domain-specific courses are those outside of computer science, statistics, and mathematics, such as in biology, economics, or psychology. In cases where it was difficult to categorize a course as computer science or statistics/mathematics, we used the identity of the home department to inform the categorization. For example, if an ambiguous course was offered by a computer science department, it was categorized as a computer science course. Such cases were rare and only affected the categorization of five or fewer courses. Given the flexibility in course choice in some programs, we recorded the minimum and maximum number of credits for each of the categories as well as the minimum and maximum total credits required for each program. The total credits for a program also included units that were not categorized into any of the three categories, such as internships and senior capstone projects.

## Statistical analyses

In all subsequent analyses, we excluded programs that were not data science majors, primarily to afford appropriate comparisons in coursework coverage and requirements. This resulted in the exclusion of Stanford University's data science minor and the University of Illinois at Urbana-Champaign's data science certificate.

To compare coverage among areas within each of the two frameworks, we first estimated an ordinal mixed-effects model, treating area as a fixed effect and the program as a random-intercept effect. We estimated separate models for each of the two frameworks. Based on the ordinal mixed-effects models, we performed post-hoc pairwise comparisons to assess significant differences between areas. All analyses were performed with the R programing language (*R Development Core Team, 2020*) with the aid of the tidyverse (*Wickham et al., 2019*) and ordinal (*Christensen, 2019*) packages. All data and R code are available at https://github.com/jcoliver/data-sci-curricula.

We evaluated the portion of each undergraduate major programs' total credits dedicated to computer science, statistics/mathematics, and domain-specific coursework. Given that there was variation in the required credits within programs, we used the midpoint between the minimum and maximum for each of the categories and total credits in subsequent analyses. For example, the University of Iowa's bachelor of science in data science required between 17 and 23 credits in computer science coursework, so we

used 20 as the expected number of computer science credits for this program. We tested two hypotheses with these data using one-tailed Student's *t*-tests:

1. Programs housed in computer science units have more required coursework in computer science than programs housed in other academic units.
2. Programs housed in statistics or mathematics units have more required coursework in statistics and mathematics than programs housed in other academic units.

### Limitations

We based program assessments solely on course titles and descriptions rather than on syllabi. Although they often provide detailed descriptions of course material, syllabi availability is highly variable, and course content at the level of syllabi may vary by term and instructor. By focusing on course descriptions, which were accessible online for all institutions and programs investigated (Table S3), we were able to consistently assess program performance. Course titles and descriptions themselves were variable within and among institutions. For example, some course descriptions consisted of a short enumeration of topics (e.g., the description for STAT 102, Data, Inferences, and Decisions at the University of California, Berkeley), while others included course content as well as format (e.g., DS 340W, Applied Data Sciences at Pennsylvania State University). There remains potential for bias among the different areas/sub-areas described in the two frameworks. That is, areas/sub-areas we characterized as generally poorly covered (see "Results") may reflect systematic poor representation in course descriptions because the topics were not mentioned in descriptions even though they may have been taught in the course. However, with only five exceptions, all areas/sub-areas scored a minimum of 3 for at least one program, indicating that nearly all areas/sub-areas could be described in a course description with enough detail to warrant the highest score possible. Finally, the focus of the current work is undergraduate degree programs offered at doctoral-granting universities in the United States; thus, care is needed in extrapolating the implications to other types of institutions (primarily undergraduate institutions, liberal arts colleges, universities outside the United States, etc.).

## RESULTS

For the guidelines set forth by the NASEM (NASEM framework), some areas/sub-areas were generally well covered in the programs we evaluated, while other areas/sub-areas received little to no attention in formal coursework (Fig. 1). The highest scoring area was data description and visualization, which primarily involves quality assessment and exploratory data analysis (mean: 3.5, median: 3.7). Most programs also paid substantial attention to computational foundations, which includes abstraction, algorithmic thinking, and programing concepts (mean: 3.4, median: 3.4). In contrast, areas/sub-areas focusing on reproducibility and ethics in data science were generally not covered in undergraduate curricula. Reproducibility, including design of workflows and reproducible analyses, was rarely indicated in course descriptions (mean: 1.6, median: 1.0). Ethics of data science, covering privacy, confidentiality, and misrepresentations of data and results,

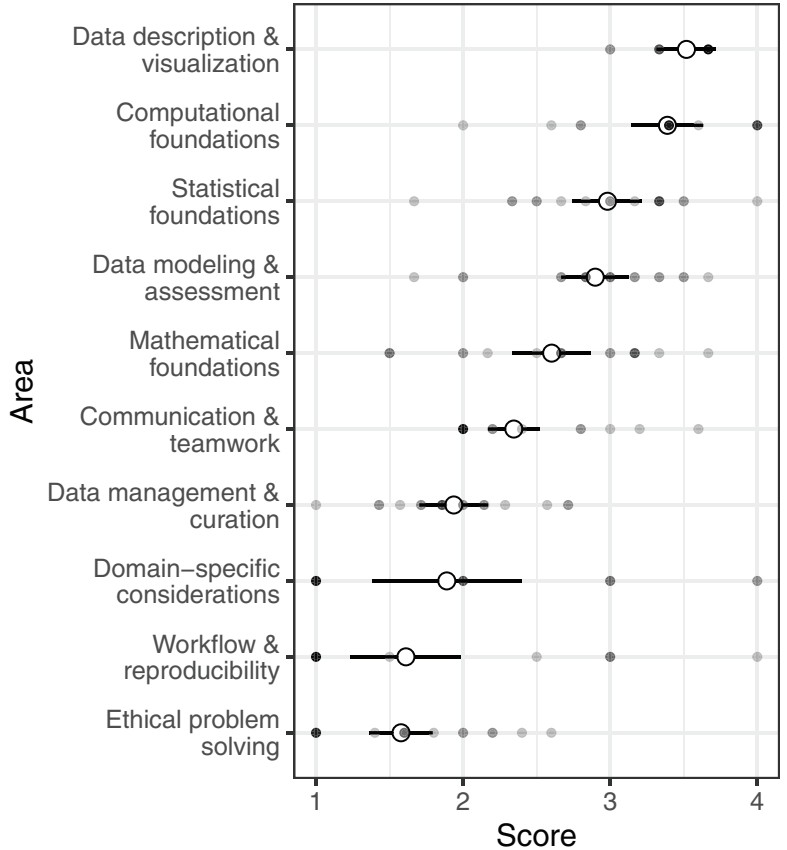

**Figure 1** **Undergraduate data science programs' scores for the NASEM framework.** Open circles show the average score for each area across all programs (±2 SE). Filled circles show mean scores in each area of the framework for individual programs.

received the lowest average score (mean: 1.6, median: 1.6). Post-hoc pairwise comparisons among areas illustrated computational foundations, statistical foundations, and data description and visualization all had significantly higher scores than reproducibility, ethics, and domain expertise (Table S4).

In regards to the GDS framework, programs scored, on average, high in all areas/ sub-areas except science about data science (mean: 1.9, median: 2.0), which is the explicit investigation of data science as a field (Fig. 2). In post-hoc pairwise comparisons among area scores, science about data science scored significantly lower than all other areas except data gathering, preparation, and exploration (Table S5).

The majority of programs investigated were characterized by coursework focused on mathematical and statistical foundations (Fig. 3). In 12 programs, courses in mathematics or statistics made up over 50% of the required coursework. In all but two programs, computer science courses accounted for less than 50% of the required coursework. Required coursework in domains outside of computer science, statistics, and mathematics was relatively low, and in only one program did domain coursework exceed 25% of required coursework. The academic unit administering the degree program significantly

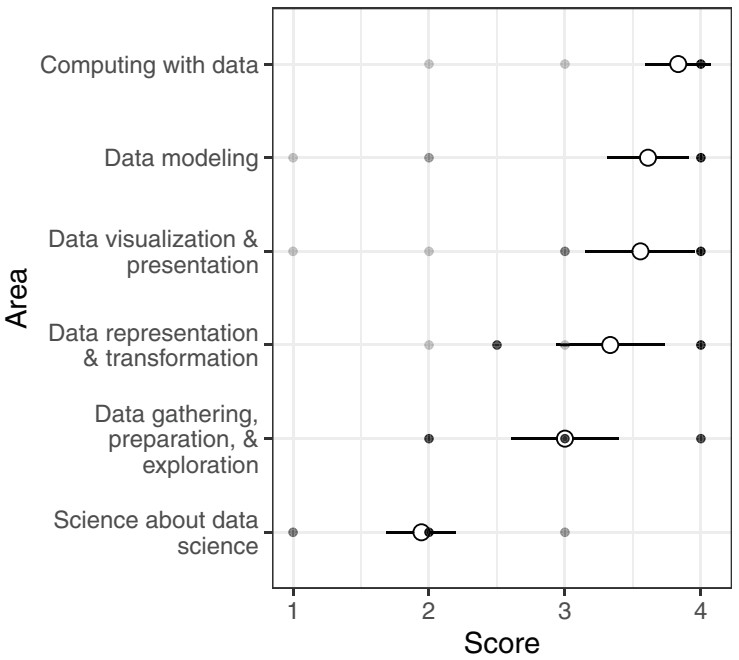

**Figure 2** **Undergraduate data science programs' scores for the GDS framework.** Open circles show the average score for each area across all programs (±2 SE). Filled circles show mean scores in each area of the framework for individual programs.                

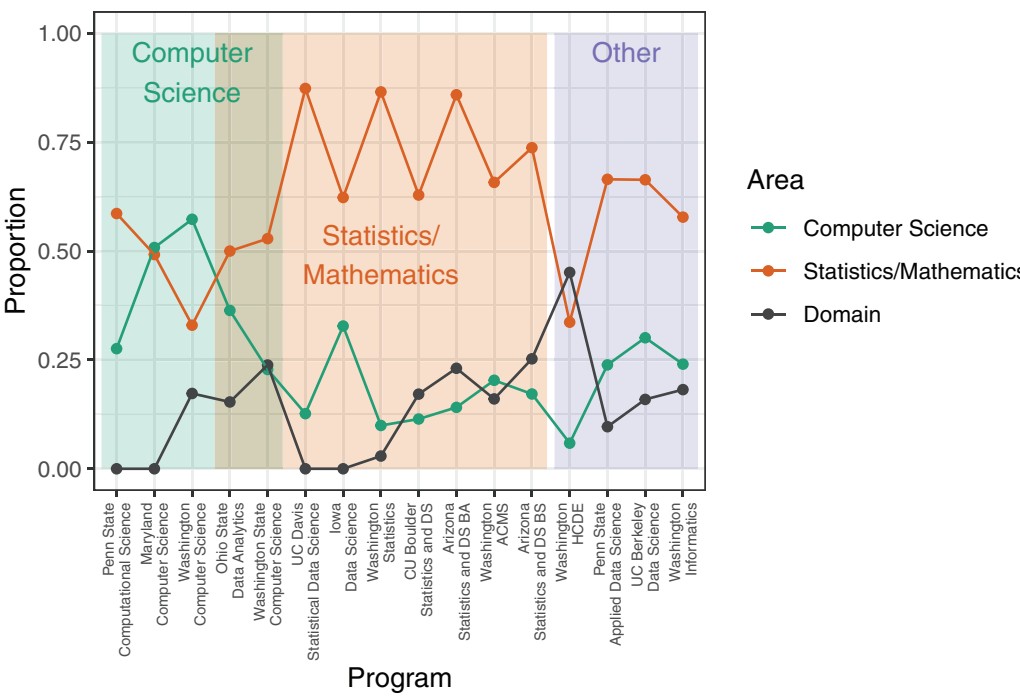

**Figure 3** **Proportion of coursework in computer science, statistics/mathematics, and domain-specific courses for each data science degree program.** Values are the proportion of total units/credits required for a degree in the data science program; see Methods for calculation details. Shaded rectangles indicate affiliation of academic unit(s) that administer the degree program.

influenced the proportion of the coursework dedicated to computer science and to statistics and mathematics. Programs administered by computer science units had significantly more required computer science courses than did programs administered by other units (mean percentage of computer science coursework in programs housed in computer science units: 39%, other units: 18%; $t = 2.899$, $p = 0.016$). Similarly, programs housed in statistics or mathematics units required more coursework in statistics and mathematics than did programs housed in other units (mean percentage of statistics/mathematics coursework in programs housed in statistics or mathematics units: 70%, other units: 52%; $t = 2.441$, $p = 0.015$).

## DISCUSSION

Our comparison of undergraduate curricula to two seminal data science frameworks reflects a focus on theoretical foundations and quantitative skills (Figs. 1 and 2). Under both frameworks, training in computational foundations was among the highest scoring areas. Indeed, computational applications addressing big, complex problems have been a hallmark of data science. Most programs also scored well in statistics and mathematics training, as well as data modeling. The emphasis on quantitative skills training, coupled with theoretical concepts underlying applications, indicates a focus on the statistical and computational underpinnings of data science and echoes similar findings in prior work on undergraduate (*Aasheim et al., 2015*) and graduate (*Tang & Sae-Lim, 2016*) data science programs. While curricula also scored well in data description and visualization (NASEM framework) and data visualization and presentation (GDS framework), the competencies described in these areas are primarily concerned with exploratory data analysis and quality assurance processes rather than using visual representations of data to communicate ideas. This statistical and computational focus is further evidenced by a heavy bias towards the number of course units in computer science, statistics, and mathematics in undergraduate data science programs (Fig. 3).

Our evaluation of programs presents a *sensu stricto* (*s.s.*) definition of data science education for undergraduates, aligning with the description of the field in the GDS framework (*Donoho, 2017*), whereby most programs emphasize the "hard skills" associated with computer science, statistics, and mathematics. Accompanying the GDS framework, *Donoho (2017)* posited that academic data science degree programs focus on statistics and machine learning, with some attention given to the technology required to compute on big data. This view of data science *s.s.* does little to include substantial training in domain knowledge outside of computer science, statistics, and mathematics. For example, the undergraduate data science curricula described by *De Veaux et al. (2017)* implies one or two domain-specific courses provide sufficient formal training in a domain. For the most part, the programs evaluated here likely equip graduates with a varied skill set for analyses and prediction, but graduates may lack the appropriate context for designing and evaluating domain-specific data science applications.

In contrast, data science *sensu lato* (*s.l.*) would include substantial training in communication, ethical considerations, and knowledge in the domain to which analyses and predictive modeling are applied. The NASEM framework called out the importance of

domain knowledge for effective application of data science, yet few programs went beyond requiring two additional courses outside of computer science, statistics, and mathematics. One notable exception is the human centered design and engineering data science option at the University of Washington, although the emphasis on domain-specific education may come at the cost of reduced training in computational skills and statistics (Fig. 3). The curriculum described in *Anderson et al. (2014)* provides another example of undergraduate training in data science *s.l.*: Students were required to take substantial coursework (15–22 units) within a "cognate," such as biomechanics, geoinformatics, or sociology. This view of data science training supports the notion that sufficient domain background is required to understand the context of models and analyses (*Provost & Fawcett, 2013*), justifying substantial consideration of domain knowledge in data science training (*Berthold, 2019*; *Irizarry, 2020*).

The degree to which data science education includes domain specialization is likely influenced by the academic affiliations of those who set the criteria. This reflects a common phenomenon within data science: The definition of data science (and by extension, data science education) depends on who is doing the defining (*Provost & Fawcett, 2013*). Just as we found that the academic unit in which the undergraduate program was housed had a significant effect on the amount of computer science and statistics coursework, the academic fields of those people creating curricula may affect how much emphasis there is on domain knowledge. For example, the undergraduate data science curricula of *De Veaux et al. (2017)* includes, at most, three domain-specific courses (one introductory, one intermediate, and one capstone), and the authors of this curriculum are all from departments of computer science, statistics, or mathematics. In contrast, the curriculum presented by *Anderson et al. (2014)* includes substantially more coursework in a domain other than computer science, statistics, or mathematics and was created by faculty from biology as well as computer science and mathematics. Such differences among curricula further illustrate data science as an evolving field and demonstrate considerable heterogeneity in what can be expected from recent graduates of undergraduate data science programs.

Similar to the paucity of attention to domain knowledge, most programs did not explicitly provide training in workflows, reproducibility practices, and the ethics of data use and reuse. The dearth of training dedicated to ethical problem-solving is similar to earlier comparative findings (*Aasheim et al., 2015*). Best practices in reproducibility and ethics are critical for maintaining quality of data science applications (*Saltz, Dewar & Heckman, 2018*), and their omission from undergraduate data science programs potentially creates a *Promethean* workforce prepared to use a variety of computational and statistical tools in socially inappropriate ways. A growing body of examples illustrate bias in data science applications (*O'Neil, 2017*); such biases have real-world impact in criminal justice (*Lum & Isaac, 2016*), employment (*Dastin, 2018*), and healthcare (*Obermeyer et al., 2019*). These impacts further reinforce the necessity of appropriate training in ethical considerations in data science.

Areas marked by deficiencies in the two frameworks may also reflect the relative youth of the field of data science. For example, the GDS framework includes the area science

about data science, which received relatively low scores; one contributing factor could be that there remains discussion of what data science actually entails, precluding a formal study and circumscription of the field (*Donoho, 2017*; *Irizarry, 2020*). Similarly, the lack of attention to ethical considerations in undergraduate data science programs could be a consequence of the recent rise of data science. For example, the qualitative and quantitative changes to analyses and predictions brought on by the big data revolution have created a new landscape for ethical considerations, and training in ethical issues in data science remains a growth area (*Saltz, Dewar & Heckman, 2018*). In contrast to "traditional" data science topics, such as linear algebra, which have long been recognized as important for statistical analyses, ethical precepts of data science may take more time to become integrated into undergraduate data science curricula. The low scores in the ethical problem solving area of the NASEM framework may also be due, in part, to our means of assessment. Course descriptions rarely went beyond mentioning the "ethics of data science," while the sub-areas described in the NASEM framework, and thus our evaluation rubric, included specifics such as "the ability to identify 'junk' science" and "the ability to detect algorithmic bias." If these important ethical topics are included in data science curricula, course descriptions would do well to call them out explicitly.

## CONCLUSIONS

Our assessments of undergraduate data science curricula demonstrate a focus on theoretical foundations and quantitative skills with relatively little preparation in domains outside of computer science, statistics, and mathematics. This generally aligns with the "greater data science" definition provided by *Donoho (2017)*. The work here suggests that data science undergraduate students receive training similar to those enrolled in a statistics program, although additional work formally comparing statistics degree programs to data science degree programs is needed. Additionally, an evaluation of the factors influencing data science curricula is beyond the scope of this work, but future work should consider the possibility that current data science programs are reflexive responses to market demands *à la* academic capitalism (*Slaughter & Leslie, 1997*; *Slaughter & Rhoades, 2004*) or isomorphic processes (*DiMaggio & Powell, 1983*).

Many programs fell short of guidelines put forth by the *National Academies of Sciences, Engineering & Medicine (2018)*. One possibility is that the areas that were not well covered (e.g., reproducibility, ethics, domain knowledge) are not recognized by the data science community as warranting substantial training. Alternatively, it remains an open question of how realistic it is to expect an undergraduate program to effectively cover all areas described in the NASEM framework. While our assessments generally treated all competencies with equal weight, the NASEM's recommendations may afford interpretations whereby some sub-areas require significant coursework, such as a year of linear algebra, while other sub-areas merit one or a few class periods, such as ethical consideration of data science. Our reliance on course titles and course descriptions may bias against topics that receive minimal, albeit potentially impactful, coverage in curricula. Future work comparing curricular content through in-depth program reviews or exhaustive syllabi sampling could demonstrate more nuance in the variation among

undergraduate data science degree programs. Additional revisions to undergraduate data science education guidelines, including required levels of competencies, could prove helpful in defining what should be expected from a recent graduate of an undergraduate data science program.

## ACKNOWLEDGEMENTS

We thank Harry Hochheiser and two anonymous reviewers for insightful feedback on an earlier version of this work.

### Funding

The authors received no funding for this work.

### Competing Interests

The authors declare that they have no competing interests.

### Author Contributions

- Jeffrey C. Oliver conceived and designed the experiments, performed the experiments, analyzed the data, performed the computation work, prepared figures and/or tables, authored or reviewed drafts of the paper, and approved the final draft.
- Torbet McNeil conceived and designed the experiments, performed the experiments, prepared figures and/or tables, authored or reviewed drafts of the paper, and approved the final draft.

### Data Availability

Data and code are available at GitHub (https://github.com/jcoliver/data-sci-curricula) and Zenodo:

Jeff Oliver. (2020, December 16). jcoliver/data-sci-curricula: Post review release (Version v0.91). Zenodo. DOI 10.5281/zenodo.4329722.

### Supplemental Information

Supplemental information for this article can be found online at http://dx.doi.org/10.7717/peerj-cs.441#supplemental-information.

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
