# Peer review of "Undergraduate data science degrees emphasize computer science and statistics but fall short in ethics training and domain-specific context"

_PeerJ Computer Science, doi:10.7717/peerj-cs.441_

## Round 0.1 · original submission · Major Revisions

The reviewers noted some strengths of this paper, including availability of data, clarity of organization, and the quality of the analysis. Some reviewer comments deserve particular attention:

1. Both reviewers point out the limitations of relying on course names and descriptions. An exploration of the similarly between courses with similar names - or perhaps the overlap between courses with different names - might clarify the utility of this approach. Although such an analysis might be impractical for the whole dataset, an exploration of a few courses might help.

2. Reviewer 1 raises some valid concerns regarding the choice of comparator institutions. A bit more explanation here would be helpful.

2. There are some questions about the availability of some of the data (Reviewer 2)

3. Reviewer 2 also some useful suggestions for enhancing the discussion of ethics education.

Reviewer 1 ·

Basic reporting

Literature and references are well laid out.

I would suggest a more in-depth discussion about the differing definitions of data science. While it is true a full accounting of the term is beyond the scope of this paper, a single definition is not sufficient to represent this issue. I am specifically referencing "An exhaustive history of the term
and definitions is beyond the scope of this work, but we adhere to the following general
description: Data science draws on statistics and computer science to address questions in
various domains, such as biology, education, physics, business, linguistics, or medicine". I think what is missing is data science vs analytics vs data engineering. Nothing too deep, but given the nature of this particular field there should be more discussion.

Experimental design

I do not understand the reasoning behind "chose institutions comparable to our home institution". It seems completely reasonable to apply the frameworks to institutions that are outside this definition. It could even be more useful and important work to broader this. It seems like the build up to this point is somewhat misleading since the title, abstract, etc do not draw notice to this point. I think there needs to be clarity on what this paper is proposing to study from the beginning if we maintain that the only schools in consideration are under this selection bias.

Looking at course titles and descriptions is weakness of the approach it is mentioned in the paper. While not possible for all of the programs, I would suggest that scoring select programs where additional information is available and comparing that against what is determined using only titles and descriptions would be valuable. For example, the authors should reach out to contacts at some subset of these programs and ask for help reviewing their program. This could be done in a manner such that it informs how consistent the scoring is done before and after this information is available.

Validity of the findings

Please see previous comments that I believe touch on the validity or at least in inferences we can draw from the analysis.

Additional comments

The study was well conducted and the paper is well organized. My main comment (and it is what I've tried to communicate in the specifics above) is that I am not sure how much of the analysis is accurate given the limitations on titles and descriptions. I feel there needs to be more work done to establish some understanding of this measurement uncertainty.

Reviewer 2 ·

Basic reporting

Comments about the data and code

- I commend the authors for depositing data and code on Zenodo! Bravo! I appreciate that there is a readme and a license. Some additional comments to make this useful. Please add additional information the readme, including descriptions of code files, what order in which they should be run. If possible, include a Rmarkdown file that calls the appropriate scripts and runs the analyses. You can show code blocks that would be useful to most readers (like the model fitting) but hide the data cleaning code blocks (allowing an interested reader to examine those separately).

- A general comment unrelated to the manuscript but to this code deposit. Do not include rm(list=ls()) at the top of each document. That's an outdated practice to clean the current working environment. Since you are using Rstudio projects, these should run in their own environments and not be contaminated by other projects. Don't use stop statements at the top of scripts.
- For the framework na, use another name in the code as na will cause problems in your code.
- Load the packages you'll use at the top of scripts.
- The code did not run right off the bat for me. I stopped fixing bugs to focus on the rest of the paper, but share the code with another R person who doesn't have access to your machine and see if they can run it.

Experimental design

no comment here but see general comments

Validity of the findings

no comment here but see general comments

Additional comments

J. Oliver and T. McNeil tackle an interesting question in this paper. Given the rapid proliferation of undergraduate programs in data science, what do they actually teach, are they comparable, and how do they compare against frameworks designed to evaluate them. As a member of one of the institutions described in the study, I was keen to learn insights into what is working and what's missing.

Very early in the background, after describing the growth of data science programs (13 to 50) and the lack of clear definitions, the authors are quick to point out that the programs lack coursework on ethical use of data/comms skills. This seemed very abrupt and additional background tying this back to the frameworks (NASEM) would be helpful. If the goal of the paper is to also demonstrate how the programs are not uniform, comparable, and heavily influenced by the departments running them (stats versus cs), it would also be helpful to cite a few examples of why lack of ethics training in data science has been problematic, especially in the context of workforce development. There are many to choose from (see weapons of math description for a general reference) but a few here would be helpful.

I appreciate that there is a limitations section in the methods, given that you relied solely on course titles and descriptions. I would appreciate more acknowledgement of this in the discussion. For example, reproducibility is rarely taught as a separate course and the appetite for such courses is also currently limited even among graduate students/postdocs. I am aware of many data science courses at several of the institutions you have surveyed that incorporate elements of reproducibility in data science courses. Data science ethics may also be listed as discussion topics and not show up in general course descriptions. I believe it would be important to acknowledge this. A small recommendation is that you could suggest that a future study could topic model syllabi to assess against the two frameworks.

There is no mention of the tradeoff in covering domain specific skills while trying to build up foundational computing/statistical skills. The discussion covers a notable exception being UW HCI program. The dataset doesn't include the list of courses and descriptions before they were scored. Are those also available?

While I agree that it is critical to teach reproducibility at the undergraduate level, and their omission has cascading effects, it would be fair to say that those topics are more likely to be taught at the graduate level, where students apply these skills to domain problems. Reproducibility is already a hard sell. So it is not that the data science community doesn’t consider them critical for training, it just remains unrealistic for UG programs to cover everything.

I appreciate you acknowledgement that it is early days for data science undergraduate programs so it's not surprising that they all fall short of the NASEM guidelines.

---

## Round 0.2 · accepted · Accept

Both reviewers were satisfied with your revisions and agreed that this paper is suitable for publication. Thank you for your careful attention to the constructive reviews.

Reviewer 1 ·

Basic reporting

No improvement necessary for publication.

Experimental design

No improvement necessary for publication.

Validity of the findings

No improvement necessary for publication.

Additional comments

No improvement necessary for publication.

Reviewer 2 ·

Basic reporting

Nothing further to add

Experimental design

Nothing further to add

Validity of the findings

Nothing further to add